# Durable Antimicrobial Behaviour from Silver-Graphene Coated Medical Textile Composites

**DOI:** 10.3390/polym11122000

**Published:** 2019-12-03

**Authors:** Nuruzzaman Noor, Suhas Mutalik, Muhammad Waseem Younas, Cheuk Ying Chan, Suman Thakur, Faming Wang, Mian Zhi Yao, Qianqian Mou, Polly Hang-mei Leung

**Affiliations:** 1Materials Synthesis and Processing Lab, Institute of Textiles and Clothing, The Hong Kong Polytechnic University, Hung Hom, Kowloon, Hong Kong SAR 999077, China; suhas.mutalik@polyu.edu.hk (S.M.); muhammad.wa.younas@polyu.edu.hk (M.W.Y.); cheuk-ying.chan@connect.polyu.hk (C.Y.C.); suman.thakur@polyu.edu.hk (S.T.); faming.wang@polyu.edu.hk (F.W.); 2Department of Health Technology and Informatics, The Hong Kong Polytechnic University, Lee Shau Kee Building, Hung Hom, Kowloon, Hong Kong SAR 999077, China; mian-zhi.yao@connect.polyu.hk (M.Z.Y.); qianqian.mou@connect.polyu.hk (Q.M.); polly.hm.leung@polyu.edu.hk (P.H.-m.L.)

**Keywords:** in situ reduction, *E. coli*, silver nanoparticle, polyviscose, reduced graphene oxide

## Abstract

Silver nanoparticle (AgNP) and AgNP/reduced graphene oxide (rGO) nanocomposite impregnated medical grade polyviscose textile pads were formed using a facile, surface-mediated wet chemical solution-dipping process, without further annealing. Surfaces were sequentially treated in situ with a sodium borohydride (NaBH_4_) reducing agent, prior to formation, deposition, and fixation of Ag nanostructures and/or rGO nanosheets throughout porous non-woven (i.e., randomly interwoven) fibrous scaffolds. There was no need for stabilising agent use. The surface morphology of the treated fabrics and the reaction mechanism were characterised by Fourier transform infrared (FTIR) spectra, ultraviolet-visible (UV–Vis) absorption spectra, X-ray diffraction (XRD), Raman spectroscopy, dynamic light scattering (DLS) energy-dispersive X-ray analysis (EDS), and scanning electron microscopic (SEM). XRD and EDS confirmed the presence of pure-phase metallic silver. Variation of reducing agent concentration allowed control over characteristic plasmon absorption of AgNP while SEM imaging, EDS, and DLS confirmed the presence of and dispersion of Ag particles, with smaller agglomerates existing with concurrent rGO use, which also coincided with enhanced AgNP loading. The composites demonstrated potent antimicrobial activity against the clinically relevant gram-negative *Escherichia coli* (a key causative bacterial agent of healthcare-associated infections; HAIs). The best antibacterial rate achieved for treated substrates was 100% with only a slight decrease (to 90.1%) after 12 equivalent laundering cycles of standard washing. Investigation of silver ion release behaviours through inductively coupled plasmon optical emission spectroscopy (ICP-OES) and laundering durability tests showed that AgNP adhesion was aided by the presence of the rGO host matrix allowing for robust immobilisation of silver nanostructures with relatively high stability, which offered a rapid, convenient, scalable route to conformal NP–decorated and nanocomposite soft matter coatings.

## 1. Introduction

The alarming rise and spread of antibiotic drug-resistant infections (often referred to as superbugs) worldwide have been regularly identified as among the major contemporary global health risks by both the World Health Organization and the World Economic Forum [1,2]. Superbugs arise from the overuse and abuse of antibacterial drugs. Antimicrobial resistance (AMR) grows with overexposure. For example, aggressive and long-lived bacteria such as methicillin-resistant *Staphylococcus aureus* (MRSA; a superbug that can survive 180–360 days depending on the environment and contact surface) have emerged rapidly and spread globally in recent years. Other multi-drug resistant bacteria include *Escherichia coli* (*E. coli*), a foodborne pathogen that may cause haemorrhagic diarrhoea, kidney failure, pneumonia, and more [3,4,5,6]. With continued rapid urbanisation and swelling worldwide population, common illnesses could once again become deadly, especially in densely populated cities warmer and in more humid parts of the world (e.g., Hong Kong, etc.). If left unaddressed, then, under such conditions, organisms can readily spread among the general population and in hospitals.

Nosocomial infections (such as healthcare-associated infections (HAI)), are absent at the time of hospital admission, but acquired and present ≥48 h after admission [7,8,9,10]. HAI affect ~10% of all admitted patients with a ~1% mortality rate and a ~3% mortality contribution to other diseases, which amounts to millions of unnecessary deaths worldwide [11]. HAI causes devastating economic and capacity utilisation problems while treatment options are extremely limited, partly because vulnerable affected subjects are often already immune-compromised in hospital settings and, subsequently, encounter resistant micro-organisms. Coupled with the increasing complexity of (often invasive) contemporary medical interventions, the result can be heavily prolonged recovery (lengthier hospitalisation by 5–10 days on average, with significant rises in associated healthcare costs), in the best case [11,12,13,14]. Thus, in addition to reducing use of known antibiotics, alternative antibacterial agents with robust and multi-action mechanisms are urgently required [15].

Medical textiles have multiple functions, including absorption of bodily fluids and the promotion of wound healing. In healthcare, the most utilised type of multi-use textiles are cellulosics, polyesters, and their blends (e.g., polyviscose), usually in the non-woven form. Non-wovens have a large surface area, high porosity, good permeability, and are easily processed [16]. Such benefits have led to wide-ranging uses, e.g., diapers, sanitary napkin covers, hospital gowns, surgical drapes, sheets, pillow cases, bedspreads, and more. However, many such conventional textiles act as bacterial exposure routes to the outside world because the microscopic crevices found in fibers allow micro-organism incubation, which eventually spread infection to patients and medical staff, despite rigid hygiene protocols. Examples include HAI of non-bloodstream related, endogenous origin occurring mainly due to contact with hospital gowns and sheets, and polyviscose surgical drapes that are meant to isolate patients from operating theatre staff and the environment during procedures, which often allow liquids to pass through, acting as a potential source of bacterial contamination to patients in an already highly compromised state. As such, antimicrobial properties in medical textiles are highly desirable [17].

Surface treatment routes towards long-lasting, passively-acting antibacterial medical textiles could be clinically and economically effective in disrupting bacterial transmission cycles [18,19,20,21]. A version of these fabrics are already used in healthcare environments with high reported efficacy [22]. As such, techniques based on functionalised medical textiles, that are essentially self-sterilising, are now areas of focus, including various routes towards nanoparticles (NP) incorporated into polymers and coatings [23,24,25,26,27,28]. Silver (Ag) and copper (Cu) compounds are among the most studied metallo-pharmaceuticals, with textiles, implants, catheters, etc., currently using Ag-species and engineered delivery systems involving Ag-derivatives, as antimicrobial agents against *E. coli*, *S. aureus*, etc. While there are certain cases where nanomaterial use is restricted in medical textiles (and compounds such as N-halamines dominate instead) [29], NPs are already established across a vast range of medical-related products. Lately, the focus has been combatting multi-drug resistant bacteria and broader applications in healthcare settings, such as wound dressings and especially in controlling biofilm infections, which imparts a self-sterilising functionality to medical textiles [30,31,32,33,34,35].

A major problem for washable and re-usable functional substrates is that water-soluble AgNP cannot usually be firmly deposited on textiles due to poor wash fastness. The significant loss of AgNP from a substrate upon laundering (more so when exposed to strong oxidising agents), even though leaching extent is substrate-dependent [36]. Another problem with bound antimicrobials is potential abrasion from textile fibres or even deactivation, i.e., long-term durability and functional performance issues. These sloughing effects raise concerns about potential leaching, entry into aquatic food chains, and extended human toxicity dangers. Thus, while nanotechnology-based treatments can greatly reduce a microbial bio-burden, the potential cytotoxicity issues, compounded by the relatively high Ag-concentrations needed to reduce infection, which means leach minimisation methods are currently of great academic and clinical interest [37,38].

2D materials such as graphene-based structures have great potential in wound care and management [39,40,41,42]. Reduced graphene oxide (rGO) is an oxidised product of graphite that affords a large specific surface area and comprises several layers (usually <8) that could potentially be used as a shielding and trapping agent to increase loading of AgNP and ensure less ready release, since the electron density of its aromatic rings is substantial enough to repel atoms and molecules trying to pass through its honeycombed ringed structure [43,44]. This is aided by the broad solution-processability of the GO precursor, with conversion to rGO requiring a high activation barrier be surmounted, which is normally achieved by chemical reduction or various heating methods [45]. Unlike pristine graphene, the oxidised graphene derivative possesses structured defects. A multitude of hydroxyl, carbonyl, carboxyl, ether, and epoxy groups that provide numerous attachment points, along the basal planes and edges of the structure, such that AgNPs can be stably dispersed throughout a substrate [46,47]. Thus, use of rGO, with lateral widths in the micrometre range, acting as anchoring sites for AgNPs, could be used as a protective matrix to enhance loading and improve stability, so as to help control leaching of AgNPs, while the large number of oxygen-containing functional groups present an increase the adsorption capacity for metal ions, via strong complexation or cation exchange processes [47,48,49]. rGO has recently been used to sequester metal NP to obtain materials with multi-faceted applications, while the encapsulation of NP within polymer substrates is thought to reduce NP-loss by washing or wiping the surface [50,51]. Such systems based on plasmonic nanostructures and graphene are known as ‘plasmon-graphene hybrids’ [52].

This article reports on the durable-incorporation of anti-microbially active AgNP species into polyviscose medical textile substrates, via a simple wet-chemical surface treatment process, which is all completed at room temperature and under ambient conditions. Treated substrates displayed high air stability and chemical inertia, via a reduced graphene oxide (rGO) protective composite, i.e., the formation of AgNP-rGO composites within substrates for increased loading and enhanced active coating longevity [53,54]. The performance of these composites was compared against solely AgNP- and rGO-impregnated fabrics, with the presence of rGO acting to stabilise and facilitate the loading and adhesion of AgNP nanostructures, which provides them with an improved specific surface area and subsequent improved performance [55,56]. Furthermore, leaching tests indicate enhanced and prolonged adhesion such that, even after accelerated washing cycles, the novel composites still showed high silver loading and high bactericidal efficacy over six hours of treatment, for the optimised composites. Thus, these treated textiles could impart anti-bacterial characteristics to a broad range of medical textiles, which significantly reduces cross-contamination in order to lower morbidity and mortality rates as well as yield parallel reductions in hospitalisation costs.

## 2. Materials and Methods

### 2.1. Chemicals and Substrates

*Medicom Safe Gauze Swab* (sterile, non-woven) substrates, comprising 70% viscose-30% polyester, were used. The highly porous matrix of the non-woven polyviscose fabric structure, comprising wrinkled fibers, allows a higher surface area for the adsorption of more silver precursor, and, in turn, acts as a template for AgNP synthesis [57]. Chemicals used as-received in the synthesis were ethanol (EtOH, 95%, VWR, Shanghai, China), graphite (Acros), hydrochloric acid (HCl, Sigma Aldrich Shanghai, China, 37%), hydrogen peroxide (H_2_O_2_, 30%, AWR, Shanghai, China), nitric acid (HNO_3_, Sigma Aldrich, 65%), silver nitrate (AgNO_3_, ILA, 98.8+%), sodium nitrite (NaNO_2_, Sigma Aldrich, Shanghai, China), sulfuric acid (H_2_SO_4_, 95%, VWR, Shanghai, China), and deionised water (18 MΩ), in all cases.

### 2.2. Hummer’s Method for Reduced Graphene Oxide (rGO) Synthesis

Natural graphite flakes (3 g) were added to c.H_2_SO_4_/HNO_3_ (9:1) and NaNO_3_ under stirring for 4 h at RTP. Slow KMnO_4_ (18 g) addition at 0 °C, was followed by 12 h stirring at 90 °C, and then undergoes a slow reaction with 10% H_2_O_2_ for the cooled mixture. After centrifugation, the resultant solid was washed successively with deionized (DI)-H_2_O, 30% HCl, EtOH, and then multiple DI-H_2_O cycles until the supernatant ran clear of AgCl precipitate. The solid was then filtered and dried overnight before re-dispersion in DI-H_2_O via ultrasonication at RTP for 30 min to form a golden-brown suspension. Although longer sonication can improve dispersion, the process damages the GO product. Therefore, 30 min was chosen [58]. The GO (5 mg/mL concentration) was used in the next phase of substrate impregnation and reduction, without further processing. The negatively charged, chemically-synthesised GO possesses a multitude of available functional groups on the nanosheet surface, including carbonyl, carboxyl, epoxide, and hydroxyl groups on the basal planes, which allow silver cations to interact through physisorption, electrostatic binding, or charge-transfer interactions, in addition to aiding attachment to the substrate surface [59,60].

### 2.3. Wet Chemical Experimental Impregnation Method

This study utilises an in situ impregnation and co-reduction process, based on controlled, solution-based dipping using well-mixed precursors and the variation of the NaBH_4(aq)_ reducing agent concentrations [61,62]. NaBH_4(aq)_ tends to be colour safe when used at a proper strength, and treatment on textiles can be so benign that many museums have utilised it in the cleaning and restoration of historical cellulosic textiles [63,64,65,66]. The impregnation process involves two steps: 1) fabric surface pre-treatment with NaBH_4(aq)_ for 30 min, to create defects and binding sites. 2) After removal of excess NaBH_4(aq)_ via padding, the samples were immersed in a graphene oxide (GO), AgNO_3(aq)_ or AgNO_3(aq)_–GO mixture for about 24 h in a second dip processing step. AgNPs were synthesised in situ from AgNO_3(aq)_ reduced to zero-valent species, while GO was converted to rGO using the same process. While AgNP formation using NaBH_4_ is known to have extremely fast kinetics (i.e., millisecond timescale) [67,68,69], the reduction and incorporation of rGO is a lengthier process, which necessitates a longer overall synthesis process [70,71,72]. After synthesis, samples were removed, padded of excess liquor, washed with water until it ran clear, rinsed with ethanol, and dried overnight. Specifically, with regards to the surface-mediated AgNP synthesis process, the functional groups on the surface treated textile interact with silver species and preferentially form AgNP at the surface. Thus, the process indirectly controls AgNP incorporation and size through mediation of reducing agent concentration (1–200 mmol), with a constant 0.2M AgNO_3(aq)_ precursor solution concentration, into which the surface-treated substrate is submerged. Because textile surfaces are irregular and comprise various pores and voids, such areas act as effective micro-reactors that incubate growth of AgNP clusters, as well as intrinsically restricting the overall size of clusters, which effectively acts as physical capping agents such that chemical additives are not required [59,60,73]. The entire coating/deposition process occurs at room temperature and pressure. No high temperature/low-pressure processing is required, except for the drying stage (~60 °C, O/N).

Only single-dip sequences of reducing agent impregnation–active species incorporation for Ag- and rGO-assimilation were carried out because chemically prepared GO sheets tend to possess carboxylic acid, epoxide, and hydroxyl functional groups that can act as repulsive forces when attempting to form multi-layered systems based on GO [74]. During the synthesis, when the GO dispersion is added to the AgNO_3(aq)_ solution, a precipitate formed over time, which, due to electrostatic attraction, leads to the attachment of cationic Ag particles on the anionic GO surface, with the resultant composites precipitating from the solution. Nevertheless, incorporation and divergent effects due to composite incorporation were clearly observed on the fabric substrates.

### 2.4. Antimicrobial Testing (AATCC TM100; ISO20743)

Antimicrobial activities of composite materials were screened in vitro against the common reference pathogen, *Escherichia coli* American Type Culture Collection (ATCC) 25922, which is a gram negative bacterial strain, under aseptic conditions throughout [75,76]. In order to generate reliable and reproducible results, the standard ‘lawn culture’ method was used, where an inoculum is spread over an agar plate using the same inoculation dosage and incubation time, such that all samples are measurable and undergo consistent operational conditions. Bacterial growth was measured by the counted colony forming units (CFU) measuring bacterial growth on agar plates [77]. Experiments were conducted in triplicate and repeated on three separate occasions, both before and after washing, on fresh samples each time. The mean value with a standard error reported in the final results. A fresh *E. coli* culture was grown on a tryptic soy agar plate (Oxoid) and a single bacterial colony inoculated in 10 mL tryptone soya broth (Oxoid) at 37 °C for 24 h with shaking at 250 rpm, prior to harvesting. The bacterial suspension was diluted 1000-fold to obtain an inoculum of ~10^6^ CFU/mL. The inoculum was confirmed by plating 10-fold serial dilutions on tryptone soya agar for viable counts. Likewise, each textile substrate swatch sample (~1 × 1 cm) was inoculated with 25 μL of bacterial suspension at a concentration of 10^5^ CFU/mL and incubated at 37 ± 1 °C for 6 h. After incubation, 450 μL saline was added to the inoculated swatch samples, which was followed by vortexing to release the bacterial cells from the substrate swatch. The contents were then diluted to 10^6^ CFU/mL, and plated (100 μL) evenly onto tryptic soy agar plates. Total bacterial count per sample was manually recorded after incubation at 37 ± 1 °C for 24 h in the dark. Log reduction of viable bacterial counts and percentage-kill for swatches with different composite materials were subsequently calculated and compared against an untreated (control) sample. The Mann-Whitney U test (a nonparametric test often used to rate pharmaceutical drug efficacy) was used to determine the statistical significance of counted antibacterial results (i.e., (i) an untreated ‘control’ blank substrate sample vs. inoculum, (ii) blank substrate vs. rGO, (iii) blank substrate vs. AgNP, and (iv) blank substrate vs. Ag-rGO), using the *R* software package.

### 2.5. Physico-Chemical Nanoparticle (NP) Characterisation

NP properties depend on their size, shape, and agglomeration as well as a local physical and chemical environment and detailed characterisations that were carried out to probe these changes. UV–Visible absorbance data was acquired on a Varian Cary 300 Conc UV–Vis Spectrophotometer (Palo Alto, CA, USA), over a 200–800 nm range at a step size of 0.5 nm. Raman spectroscopy data was acquired on a BaySpec Nomadic Raman Microscope (San Jose, CA, USA) using a 785 nm laser excitation source, over the 200–3200 cm^−1^ range at RTP (room temperature and pressure). Attenuated total reflectance Fourier transform infra-red (ATR-FTIR) spectroscopy was acquired at a Perkin-Elmer Spectrum 100 (Waltham, MA, USA), over the 650–4000 cm^−1^ wavenumber range in the transmittance mode, with 0.2 cm/sec scan speed and 20 scans. The polymer composite degradation properties were mapped using thermogravimetric analysis (TGA) in triplicate and differential scanning calorimetry (DSC) cycling tests on a Mettler-Toledo Star e TGA Thermogravimetric Analyzer (Columbus, OH, USA), under N_2_ (20 mL/min) conditions in dynamic mode, at a 10 °C/min ramp rate, over the 50–500 °C range. Scanning electron microscopy (SEM) was done on a Field Emission Electron Microscope JEOL JEM-2100F (Akishima-shi, Japan) at accelerating voltages up to 20 kV with fabric samples sputter coated with gold at low-vacuum. Cluster analysis was carried out using the linear intercept method, from 15 separate points at a time, on ImageJ (National Institutes of Health, Bethesda, MD, USA) [78]. Microscopic images of fabrics were taken on a Leica M165C (Wetzlar, Germany). Fibre cross-sectional and longitudinal analysis was carried out on a Nikon Opthiphot-Pol (Tokyo, Japan). Fabric thickness was tested on a Ames Thickness Testor (B.C. Ames, Framingham, MA, USA), using a modified version of the ASTM D1777-96 method, taken at 30 s under applied pressures of 4 gf/cm^2^ (0.392 kPa, 39.2 mN/cm^2^) and 30.7 gf/cm^2^ (3.012 kPa, 301.2 mN/cm^2^), for fabric samples ~6.5 × 4.5 cm, and the difference between them used to calculate the fabric surface layer thickness [79]. In each case, specimens were tested in triplicate and the average values were reported. Wide angle X-ray diffraction (WAXD) was employed to identify crystalline silver in the composite structures, and collected on a Rigaku SmartLab X-ray diffractometer (Tokyo, Japan) with Cu Kα radiation (λ = 1.542 Å), operating at 60 kV and 60 mA, over the 15–90° 2θ angle range at a step size of 0.01. Nanoparticle size and size distribution of the particles have an impact on the agglomeration behaviour of the precursor mixtures and were measured at a pH of ~7, on a Brookhaven Instruments Corporation ZetaPlus Potential Analyzer (Holtsville, NY, USA).

### 2.6. Wash Fastness of Composite Textiles (AATCC TM61-2001; ISO105-C10:2006)

Wash durability was evaluated at 40 °C with 4 g/500mL ECE Phosphate Reference Detergent (SDC Type 3) for 30 min, with 10 steel balls (each weighing ~0.9 g, with a diameter of 0.7 mm), in a Launder-O-meter (SDL Atlas, Hong Kong, China) [80]. Four accelerated wash cycles (equivalent to 12 conventional laundering cycles) were conducted on the polyviscose composites (further cycles were not possible due to substrate damage resulting from polyviscose sensitivity to laundering and drying). Antimicrobial testing and chemical characterisation of sample swatches was carried out before and after wash fastness testing.

### 2.7. NP Leachant Testing Using Inductively Coupled Plasma Optical Emission Spectroscopy (ICP-OES)

The content of Ag-leaching in the substrates was determined using an Agilent Technologies 5100 ICP-OES (Santa Clara, CA, USA). Following sample preparation, polyviscose substrates of either Ag or Ag-rGO coating, (1.0 × 1.5 × 0.15 cm, ~15–18 mg) were immersed in 20 mL saline solution (0.85%), stirring at 100 rpm for 3 h at 40 °C, and the concentration of silver released into solution measured. Silver concentration was determined against a calibration curve prepared using a Sigma Aldrich ICP silver element standard solution. The calibration curve method utilises an internal standard to minimise errors caused by instrumental drift and reduces chemical matrix effects [18,81,82]. A blank and a three-point calibration curve were generated using concentrations of 1.0, 5.0, and 10.0 ppm of the silver element standard, and a linear fit chosen in the on-board software. Sample dilution was not required since the ppm levels detected were under the calibration limits. For quantification, the calibration curve had a correlation coefficient R^2^ > 0.995, with the mid-point of the calibration curve well within 1% of the theoretical concentration. The calibration and textile samples were analysed in 0.5M HNO_3(aq)_.

## 3. Results and Discussion

### 3.1. Nanoparticle-Impregnated Polyviscose Substrate Synthesis

In this study, well-dispersed silver nanoparticles (AgNP) were immobilised over a reduced graphene oxide (rGO) support via a facile two-step wet-impregnation method at room temperature, which leads to well-defined “plasmonic-graphene hybrids” over the surface of non-woven, polyviscose substrate supports. This correlates with past reports on alternative hard supports [83,84,85]. The process is substrate-mediated. Thus, rather than use a surfactant, the rough, porous surface of the textile fabric substrate comprising a multitude of functional groups and micropores acts as the micro-reactors and nucleation points as well as size-controlling agents that constrain the size of AgNP agglomerations, which allow an effective upper band on cluster sizes [73]. The functional groups (e.g., carboxyl) are coordinated such that the silver cations directly attach via electrostatic interactions, which facilitates the co-reduction process, after having been converted to borohydride functional species [59,60]. Consequently, a stable GO-Ag nanocomposite forms. The Ag^+^/GO composite is reduced to Ag^0^ species, simultaneously with the reduction of adjacent GO to rGO [86].

While a range of reducing agent concentrations (1–200 mmol) were utilised in this project, the data presented in the main body of this article relates to the 200 mmol samples. The full datasets are available in the Appendix A. Varying NaBH_4(aq)_ concentration offers a remarkably powerful but simple route for controlled, increasingly mono-disperse and isotropic NP growth/incorporation and composite formation, with all other factors being equal. Therefore, this removes the trouble of a broader multi-component precursor variation [87,88]. Benefits include (i) minimal wastage of expensive precursors, (ii) ability to re-use unused precursor, (iii) a room temperature process, done in situ, which negates purification or transfer steps [89], (iv) controlled, increased, and uniform interaction of species throughout various substrate shapes, which is ideal for textiles and related soft matter. After coating, substrates underwent a color change. Due to deposition of AgNPs, the colourless substrates turned a yellowish brown. Those with rGO turned a grey-black while the Ag-rGO combinations yielded a brown-black colour.

### 3.2. Nanoparticle-Impregnated Composite Bonding Interactions, Materials Characterisation, and Durability Testing

SEM images, bright-field contrast optical microscopic images of fibers as well as longitudinal and cross-sectional images are presented in Figure 1. The optical microscope images indicate clear variations between the Ag-impregnated, rGO-impregnated, and Ag-rGO impregnated substrates. The darker coloured aggregates indicate the presence of additives as the result of light absorption or scatter, with the greater density of aggregates in Ag-rGO samples indicating a higher loading overall. SEM images clearly display characteristic rGO lamellar sheets, which are decorated with AgNP for Ag-rGO samples, which shows successful loading. The images indicate fibrous surfaces decorated with AgNP agglomerates of polydisperse distributions in shape and diameter, attached to rGO sheets [90,91]. It is important to note that particles observed on the textile surface via SEM comprise a collection of Ag crystallites and, thus, are much larger than the crystallite sizes estimated by the Scherrer equation [92]. The primarily spherical agglomerations observed in this case are also reflected in the UV–Vis data where there is no observable lateral plasmon shift signal.

After the laundering tests of 12 equivalent standard wash cycles, both simple visual appraisal and optical microscopy indicate that laundering causes fabric damage. This degradation takes the form of an increasingly flattened substrate as well as some structural thinning. SEM indicates a greater degree of aggregation and enhanced ball-like formation observed on the samples. Post-wash, it seems that the solely AgNP samples exhibit enhanced release, even though the metal presence is still detected and observed [93]. The seemingly greater presence of AgNP on the rGO samples indicates reduced relative outflow of AgNPs, i.e., higher laundering durability and binding [94,95].

XRD and EDS data are given in Figure 2a,b. EDS confirmed the presence of silver on polyviscose substrate surfaces with a significantly higher Ag-wt % reading detected for Ag-rGO samples than for solely AgNP samples. Post-laundering, while the amount of detected Ag decreases in all cases, naturally, the initial pre-wash relationship still holds, to a large degree. The mean values become ~0.4 wt % (AgNP, from ~14.2 wt %; which is in agreement with past reports of >90% material loss in textiles) [96] versus ~8.2 wt % (Ag-rGO, from ~34.3 wt %). Thus, the presence of rGO seems to improve both initial loading as well as longer-term adherence to the substrate. The silver loading values on cellulosics, as obtained through the surface treatment process, are much higher than past reports, which have ranged from 0.2–2.3 wt % silver on initial textile loading [97,98,99,100,101]. This highlights the efficacy and potency of the surface treatment route adopted in this study. The expected XRD signals for GO/rGO, which are usually observed at ~10° and ~25° 2θ, were completely blocked by the polyviscose substrate signal [102,103]. Nevertheless, the silver incorporated samples yielded prominent peaks that were solely assigned to the well-defined crystalline FCC phase. There was no evidence of oxidation, nitridation, or any other mixed phase presence. The observed peaks at approximately 38°, 44.5°, 64.5°, 77.5°, and 81.5°, correspond to (111), (200), (220), (311), and (222) planes, respectively. The data seemingly indicates enhanced loading of AgNP with rGO usage (as well as increasing particle size) and with increasing NaBH_4(aq)_ concentration, as reflected in intensity and peak presence. The faster reaction kinetics of AgNP formation enhancing the reduction and attachment of the graphene oxide forms to the textile substrate. Furthermore, rGO seems to confer greater interfacial oxidative stability to the AgNP, thought due to a high energy barrier presented by the diffusion of oxygen [53,72,87]. For all samples, the common trend is that thickness of the composite coating increases with increasing NaBH_4(aq)_ concentration, which leads to diminishing the substrate signal. Concurrently, for both the AgNP and Ag-rGO samples, there is an enhancement of the silver *fcc* peaks. The relative intensity and presence of the full complement of Ag peaks for Ag-rGO samples are indicative of a higher loading, which is in good agreement with EDS findings.

ICP-OES data on leaching studies (ICP-OES provides elemental specificity and high sensitivity – limits of detection in the μg/L range) was used to determine the extent of Ag-species leaching from the polyviscose substrate into the solution (Figure 2c). Use of rGO as an overlay/protective coating allowed for effective adhesion and durability of AgNP, which minimised the aggregation problem of AgNPs and allows higher overall loadings of silver to be achieved, all with long-term protection capability [59]. In both the solely AgNP and Ag-rGO samples, there was a detectable leaching of silver into the saline solution over the 180 min, even though leachant concentrations are seemingly below the values commonly observed for municipal wastewater treatment plants, and well below levels thought to induce cytotoxic effects (e.g., the US Environmental Protection Agency (EPA) sets secondary drinking water limits of 1 × 10^−6^ g.L^−1^ silver) [104,105,106]. Results indicated that small traces of silver leached into the solution up to 1.7 × 10^−7^ g·L^−1^ silver per sample for Ag-rGO and 2.3 × 10^−7^ g·L^−1^ for AgNP, respectively [107,108]. This is in contrast to past reports of leaching between 1.5–5.7 × 10^−3^ g·L^−1^ of silver from various textile samples, by Ureyen et al., under similar testing conditions. As such, these experimentally produced composites should be safe for use.

The experimental UV–Vis data for synthesized plasmonic hybrid nanostructured samples demonstrates a well-separated single absorption peak in all cases, which corresponds to the surface plasmon resonance (SPR) of spherical AgNP, as encapsulated within the polyviscose fabric substrate (Figure 3) [109]. Surface plasmons are the in-phase oscillations of free electrons at the interface of a metal and dielectric, as indicated by a strong absorption at the resonance wavelength in a UV–Visible absorption spectrum [110,111,112,113,114,115,116,117]. The SPR band position depends on parameters such as size, shape, and polydispersity of nanostructures, even though it is the ability of rGO to otherwise enhance and control the loading and embedding of AgNPs, due to improved specific surface areas thought to be the reason for the divergence in the signal between the Ag and Ag-rGO samples [56]. Resonance wavelengths strongly depend upon the refractive index of the dielectric medium. It increases if an additional thin layer of high dielectric constant material is over-coated, and decreases for a lower dielectric constant material [74]. Thus, for Ag-rGO, the synergistic plasmonic effects between AgNP and the rGO network, and the differing dielectric constants between the two components, affects the character of the SPR signal [52]. Typically, solely a silver presence is said to suffer from chemical instability that leads to oxidation and results in an SPR red-shift and dampening. Thus, silver stability to oxidation increases with rGO use, which is in accordance with past reports where, in general, protective coatings have helped retard silver corrosion, as well as alter the optical properties of silver [53].

Related to the ICP-OES data, the enhanced loading and adhesion of the Ag-rGO composites was investigated via the SPR band profiles after 12 equivalent standard washing cycles (due to sample physical damage, while further wash cycles were not conducted). Figure 3 shows that both AgNP and Ag-rGO samples still displayed strong SPR bands even though, seemingly, the use of rGO was absent. More silver leached from the substrate than when rGO was used concurrently. This, indicates that AgNP are still present in substrates after serial washing, which indicates wash-fastness to laundering of samples.

Raman spectra of the impregnated substrates, as well as spectra of the GO and rGO solid precipitates are shown in Figure 4. The GO and rGO precipitate samples (i.e., standalone and not impregnated within the fabric) displayed the characteristic D (≈1350 cm^−1^) and G (≈1590 cm^−1^) bands of graphitic materials. The D band is related to defects in the graphitic structure (i.e., sp^3^-bound carbons) and the G band signals C=C stretching vibrations of graphitic sp^2^-carbons. The intensity ratio of D to G bands (I_D_/I_G_) is used to indicate the degree of the disorder such as defects, ripples, and edges. In this case, the I_D_/I_G_ was ~0.98 for GO and ~1.23 for rGO. The increased ratio is due to a decrease in the sp^2^ domain as a result of the reduced size of GO sheets after chemical reduction, which is driven by the defects created after removal of oxygen functional groups (e.g., epoxides, the predominant formation of ketone and carboxylic acid functionalities on the edges of GO sheets, and the restoration of the graphitic sp^2^ network) [118,119,120].

For all silver-containing fabric samples, and most of the rGO containing samples, two intense overlapping Raman bands are observed at ~1325–1355 and ~1560–1590 cm^−1^, and are again, attributed to the D and G-bands, respectively. However, the relative weakness of the bands for rGO samples indicates that the amount of impregnated/coated material is insufficient for a strong signal. However, silver-containing samples display a large signal enhancement, which is due to AgNP hot spots, i.e., it is a surface enhanced Raman spectroscopy (SERS) effect, and serves to mask most other signals [121,122,123]. The non-uniform enhancement of the respective Raman bands is a feature of SERS and makes it difficult to extract further I_D_/I_G_ data. In addition, to the main D and G bands, a sharp, clear band is also observed at ~1720–1730 cm^−1^ for most samples and is mainly due to the asynchronous stretch of C=O. The easy hydration reaction between the molecules at C=O means such vibrations are easy to disturb due to the high concentration of electron–hole pairs present [124,125]. Such a signal identification is in good agreement with the ATR-FTIR data. The decreasing signal intensity observed with higher NaBH_4(aq)_ concentration indicates that bonding is facilitated, at least in part, via reduction of carbonyl groups. The C=O signal enhancement is dependent on the short distance and subsequent orientational interaction between AgNP and carbonyl groups. As such, it seems to not have been a common feature of past reports to date simply because the nature of the non-conventional experimental synthesis method utilised in this paper, whereby the reducing agent impregnates the substrate first, prior to the Ag/Ag-rGO being incorporated in situ, which allows for a unique surface configuration to be achieved that seemingly facilitates the interaction of AgNP with the surface carbonyl groups, yielding the eventual signal enhancement.

ATR-FTIR spectra (Figure 4) include readings for samples where data were collected both before and after washing cycles, to investigate the structural changes in the impregnated fabrics as well as probe the interfacial interactions. In all cases, sample spectra were dominated by signals from the polyviscose substrate, accompanied by certain weaker composite-based signals, which indicate the relatively low loading of additive in all cases. Common signals for all samples, except the solely rGO impregnated samples, are the presence of hydroxyl stretching vibrations (~3350 cm^−1^), carbonyl stretches (~1745 cm^−1^), and bands ascribed to C–O bending and stretching vibrations of hydroxyl groups (1300–1400 cm^−1^ and ~1048 cm^−1^, respectively) [126,127,128]. The broad band observed at ~1220 and ~1110 cm^−1^ were assigned to C–O–C/ether from epoxy groups of the substrate, in addition to a common O–C=O signal (1650–1750 cm^−1^) [129,130]. The solely rGO samples had heavily reduced peak intensities for signals related to O-containing functional groups after reduction, which indicates strong and efficient interaction and conversion of the GO with the previously available hydroxyl and carbonyl type binding sites. The substrate is, thus, highly saturated. A new peak at ~1548 cm^−1^ due to sp^2^-hybridised C=C (in-plane stretching), confirms the effective reduction of GO. The strong signal is perhaps indicative of the higher relative percentage of C–C bonds from rGO incorporation and overall loading into the substrate [131]. The C–O stretching and O–H deformation vibration of carboxylic groups are represented by the peak at ~1401 cm^−1^ [132]. The comparatively lower content of oxygen functional groups present indications of involvement of O–H groups in the reduction to AgNPs [133,134]. The fact that signals are not completely absent in the equivalent AgNP and Ag-rGO combinations suggest that such substrates may not be fully saturated, which indicates only a moderate reduction of Ag^+^ to AgNP, perhaps due to a less efficient chemical conversion process, or perhaps (in the case of Ag-rGO) competitive formation effects during the simultaneous transformation of GO to rGO. After accelerated laundering, all samples indicate structural changes, with greater resemblance toward the original blank substrate profile. This indicates loss and removal of the impregnating material, with the lowest effect (i.e., least signal attenuation) seemingly observed for Ag-rGO samples.

### 3.3. Antimicrobial Properties Testing

The antimicrobial efficacy of the various nanocomposite medical textiles was tested against *E. coli*. Results reveal that Ag-rGO composites exhibit enhanced antibacterial activity (near 100% suppression for 6 h treatment, against an untreated ‘control’ blank substrate sample) as compared to solely AgNP incorporated samples (98.3 %) although a significant reduction of bacterial growth was observed in both cases (Figure 5) [135]. A reduction in the numbers of viable bacteria was observed after 6 h incubation on the treated sample substrates, in the dark, with statistically significant improved activity versus the uncoated substrate (*P* < 0.05, Table 1) [19]. Absent the use of a protective coating (e.g., silica), such post-laundering performance compares favourably with past reports [96,136]. The antibacterial activities of solely rGO-incorporated samples were limited (82.5%), although some improvement was observed as compared to the untreated surface. The reason for enhanced activity remains under question but is in line with past reports of mild antimicrobial effects, with both a compromise of cellular membrane integrity (via prolonged, direct physical contact with the rGO sheets) and oxidative stress mechanisms (as mediated by the production of reactive oxygen species) thought to be potentially responsible [127,137,138]. The enhanced *E. coli* inhibition activity of Ag-rGO hybrids exhibit promise for biomedical applications, as compared to individual rGO and AgNP agents [139]. Bactericidal efficacy is dependent on absorption variations of the formed plasmonic composite systems as well as the size, shape, and type of bacteria [140,141,142]. Past reports indicate that the rGO presence in plasmonic systems can improve charge carrier generation, enhance electron transfer efficiency, and suppress electron-hole pair charge recombination, which reduces the back-transfer of electrons [142,143,144,145]. Such properties are fundamental to the generation of free radicals that effect an antibacterial function. Thus, rGO presence, both in conjunction with AgNP, as well as on its own (to a lesser degree), inhibits growth of gram negative bacteria and also allows avoidance of oxidation by air, which enhances the stability in line with previous reports [146,147]. Alternatively, the higher loading of AgNP, and/or the presence of smaller, less agglomerated AgNP, as obtained here, enhances the known antimicrobial properties of silver, inactivating bacteria via the interaction with bacterial protein and enzymatic thiol groups [148,149]. Furthermore, the longer-term isolation of AgNPs as a result of higher loadings to laundering and abrasion ensures a degree of retained antibacterial activity [80]. This is reflected in the continued bactericidal efficacy (90.1 %) for the Ag-rGO sample as compared to an untreated ‘control’ blank substrate sample. However, post-wash, the solely rGO-incorporated sample displays severe degradation in capability, perhaps due to the amelioration of the sharp edges (as observable in the SEM images), which dent the physical mechanisms of microbial retardation, while the copious amount of extra carbonaceous feedstock material present, may serve to accelerate microbial growth.

## 4. Conclusions

This study reports on the use of reduced graphene oxide as a means of increasing overall loading and increasing the durability of adhesion of silver nanoparticles (AgNP) onto soft matter, non-woven polyviscose medical textile fabric substrates. Thus, our results show that utilisation of a simple room temperature and ambient condition synthesis method allows highly efficient loading of rGO and/or AgNP into the substrates while also affording a more even and improved dispersion of AgNP throughout the surface, all without the use of any stabilising agents. The seemingly synergic effects and resultant properties arising from interactions of the graphene analogue and the AgNP are the focus of continued study, with future attempts looking to parse the contributions of the various physico-chemical processes at play.

The output material allows for multiple benefits, including enhanced antimicrobial properties (i.e., near-100% suppression of *E. coli* observed) with improved wash fastness (i.e., ~24% retention after 12 wash cycles) as compared with standard NP-incorporated fabrics (i.e., <3% retention), which means potential rinse-reuse capabilities, for an enhanced effect as a non-migrating antimicrobial agent. As such, the substrates hold promise for application across a wide range of biomedical applications. More broadly, the general wet chemical technique is flexible and can readily be extended for use in other nanocomposite films and coatings on various soft matter substrates by varying the reducing agent concentration and/or NP-precursor. This surface treatment and modification process allows for reliably high loadings of functional materials containing functional groups that can efficiently bond to a substrate, to be efficiently incorporated into polymers (including metallic nanoparticles and graphene analogues for applications spanning biocidal, wettability, catalytic, and other applications). Thus, it offers a rapid, convenient, scalable route to conformal NP–decorated and nanocomposite soft matter coatings.

## Figures and Tables

**Figure 1 polymers-11-02000-f001:**
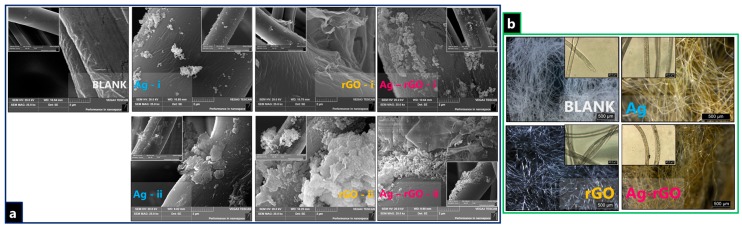
(**a**) SEM images of for nanocomposite-incorporated polyviscose fibers, both (i) before laundering, and (ii) after laundering, (**b**) Dark-field optical microscope images of nanocomposite-incorporated polyviscose fibers with longitudinal microscopy images inset.

**Figure 2 polymers-11-02000-f002:**
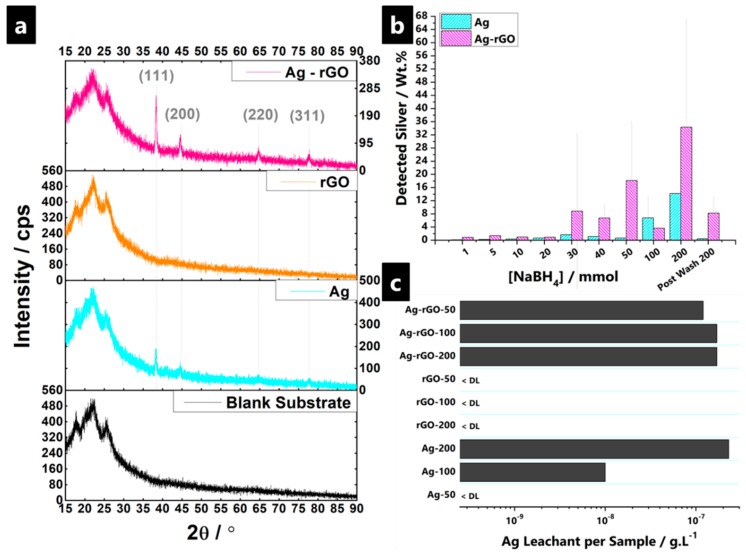
(**a**) XRD patterns of blank substrates, AgNP-, rGO-, and Ag-rGO-impregnated polyviscose non-woven fabrics, at surface treatment NaBH_4(aq)_ reduces the agent concentration of 200 mmol, which confirms the sole presence of fcc metallic silver. (**b**) EDS of fabric composites highlighting the variation in silver loading between Ag-impregnated and Ag-rGO impregnated non-woven polyviscose substrates, and (**c**) leaching of silver from Ag/rGO/Ag-rGO-impregnated polyviscose non-woven fabric into 0.85% saline solution at RTP, determined by ICP-OES, measured over 3 h. *DL: “Detection limit.”*

**Figure 3 polymers-11-02000-f003:**
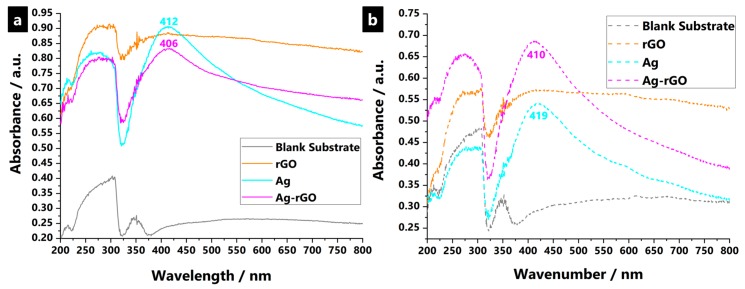
UV–Vis absorption spectra for blank substrates, AgNP-impregnated, rGO-impregnated, and Ag-rGO-impregnated polyviscose non-woven fabrics, at surface treatment NaBH_4(aq)_ reducing agent concentrations of 200 mmol, both (**a**) before, and (**b**) after, laundering durability testing, indicating the surface plasmon resonance (SPR) band where plasmonic AgNP are present.

**Figure 4 polymers-11-02000-f004:**
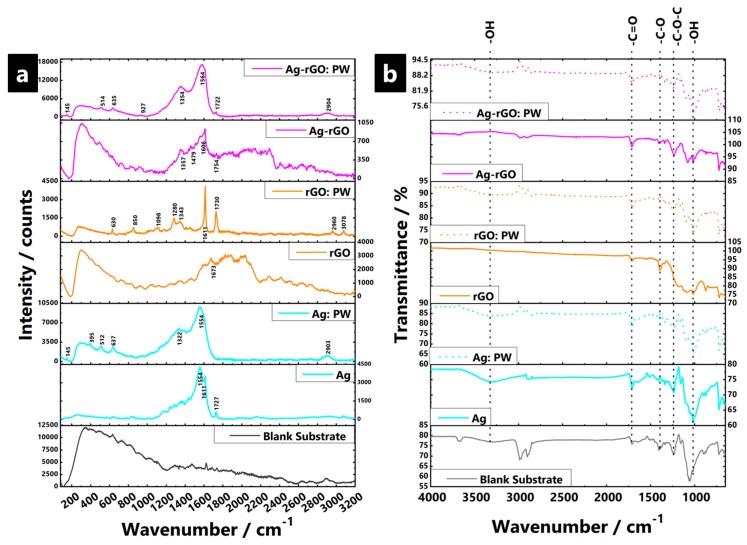
Collated Raman (**a**) and ATR-FTIR (**b**) spectra of (i) AgNP, (ii) rGO, and (iii) Ag-rGO-impregnated polyviscose non-woven fabric, at a surface treatment NaBH_4(aq)_ reducing agent concentration of 200 mmol, acquired under ambient conditions, both before and after laundering durability testing (i.e., post-wash (PW)).

**Figure 5 polymers-11-02000-f005:**
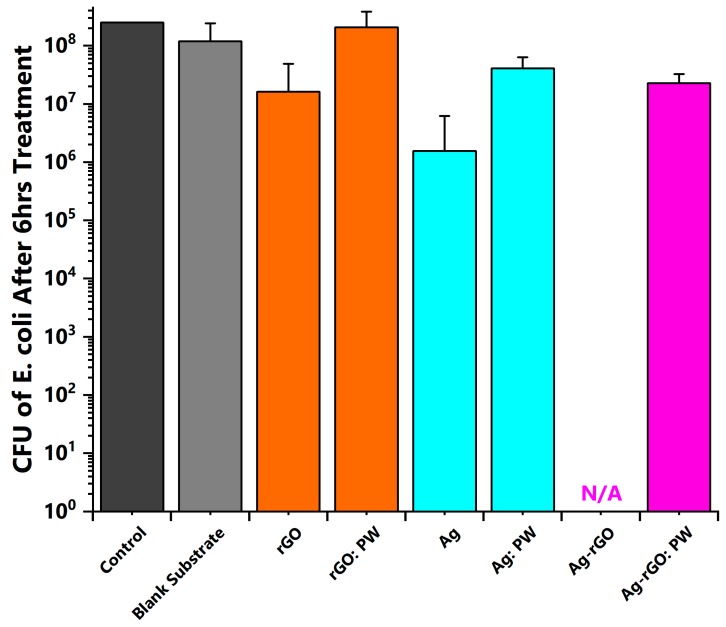
A comparison of viable *E. coli* counts after 6 h of treatment in the dark at 37 °C on modified composite non-woven polyviscose fabrics, both before and after laundering durability testing (i.e., post-wash (PW)).

**Table 1 polymers-11-02000-t001:** A comparison of the relative significance (i.e., +: significant; −: not significant) in bacterial reduction performance between the different composite systems, according to the Mann-Whitney U test, at *P* < 0.05, both before and after laundering durability testing (i.e., post-wash (PW)).

*P* < 0.05	Control	Blank Substrate	rGO	rGO: PW	Ag	Ag: PW	Ag-rGO	Ag-rGO: PW
Control	/	+	+	−	+	+	+	+
Blank Substrate	+	/	+	−	+	−	+	+
rGO	+	+	/	+	−	+	+	+
rGO: PW	−	−	+	/	+	+	+	+
Ag	+	+	−	+	/	+	−	+
Ag: PW	+	−	+	+	+	/	+	−
Ag-rGO	+	+	+	+	−	+	/	+
Ag-rGO: PW	+	+	+	+	+	−	+	/

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
