# Peer review of "Durable Antimicrobial Behaviour from Silver-Graphene Coated Medical Textile Composites"

_polymers, 2019, doi:10.3390/polym11122000_

Round 1

Reviewer 1 Report

This is an interesting paper presenting results with applicational possibilities. Taking into account a relatively low level of novelty but high practical potential I recommend to publish it after some important corrections:  

(i) "viscose" is the name of fibers made in the laboratory from cellulose. So, the name "polyviscose" is wrong and must be changed. 

(ii) The main object of modification (nonwoven) should be characterized much better, including tests of mechanical properties, surface weight, surface roughness, etc. 

(iii) some theoretical application of such modification should be proposed, this is a quite complicated modification, not for medical textiles for everyday use.  

Author Response

We thank the reviewers for their efforts and comments. We have tried to address each of the highlighted issues in turn, giving full reasoning and references, where appropriate. We hope everything is in order.

This is an interesting paper presenting results with applicational possibilities. Taking into account a relatively low level of novelty but high practical potential I recommend to publish it after some important corrections: 

(i) "viscose" is the name of fibers made in the laboratory from cellulose. So, the name "polyviscose" is wrong and must be changed.

- I don’t think this is correct. Whilst I agree that viscose is a well-known regenerated cellulose product, the material used in this paper is a blend. This was highlighted in the methodology: “Medicom Safe Gauze Swab (sterile, non-woven) substrates, comprising 70% viscose - 30% polyester, were used.” Polyester-viscose blends are commonly and widely identified by the “poly-viscose” term. Such blends, of varying percentage, are often used in, e.g.; absorbent pads, incontinence underwear, clothing etc., and have been used for many years. The presence of polyester increases the viscose wet strength, giving it the durability to handle washing and drying by machine. The blend makes the fabric stronger, but allows it to retain the same drape and feel of standard viscose. Please refer to: https://clan.com/blog/what-is-poly-viscose-fabric.

 (ii) The main object of modification (nonwoven) should be characterized much better, including tests of mechanical properties, surface weight, surface roughness, etc.

- We respectfully disagree with the recommended tests as listed by the reviewer. The reasoning is as follows:

- We believe that thickness gives a more representative effect of the effects of the coating, than does the weight. Since the coating amounts are relatively thin and the intrinsic weights of the AgNP and graphite analogues are so divergent), it would not be a meaningful data set to include. Rather, in this instance, the thickness is a better reflection of the coating’s effect on a flexible, soft matter substrate. The thickness shows clear differences (please see Fig S7b). Fabric thickness determines fabric stiffness (which in turn affects drape etc.), air permeability, moisture absorbency, abrasion resistance, etc.(please see; ISBN: 978-1-84569-372-5)

- For non-woven, soft matter substrates, the recommended exemplar for analysis is the Kawabata Evaluation System (KES). However, even KES suffers from considerable measurement issues. Indeed, this is one of the reason why many labs worldwide have often resorted to using customised characterisation methods in an effort to address this issue.  (Journal of Engineered Fibers and Fabrics, Volume 9, Issue 2 – 2014). Whilst AFM is recommended for biomaterials generally, it is seen as being deficient for textile-based soft matter fabric substrates broadly – it is only widely used to measure the roughness of a single textile filament. To the best of the author’s knowledge, AFM data has only been reported in certain cases for effective use in hydrophobic and oleophobic effects (e.g.; DOI: 10.1177/0040517513495945; DOI: 10.1177/0040517512464290, etc.). Since we are not concerned with hydrophobicity in this study (and indeed, because of the nature of the porous, non-woven, highly absorbent pads, the substrates would have little/no hydrophobic character at all), we do not think such measurements would be appropriate. Other, more involved optical techniques (e.g.; white light interferometry, focus variation) are recommended for larger-scale fabrics, with requirements of readings of the order of millimetres for a sufficient and representative overview of the textile surface. However, highly porous, gap-filled substrates are poorly suited to such methods. Furthermore, we do not have ready access to such systems at present and also feel that such readings are unnecessary to supporting the main thrust of the paper. In our opinion, the SEM and optical microscopy images gives a representative view of the various morphologies present in the fabric surface and topology, from length scales ranging from nanometre to millimetre.

(iii) some theoretical application of such modification should be proposed, this is a quite complicated modification, not for medical textiles for everyday use. 

-Agreed and added. Potential applications have been elaborated upon in the conclusions. Attention has been drawn to the value of the method and its potential in surface treatment of many other soft matter surfaces. Specifically, the seemingly higher loading potential and greater durability/longevity of subsequently output samples. The additions:

Introduction (final paragraph): This article reports on the durable-incorporation of antimicrobially active AgNP species into polyviscose medical textile substrates, via a simple wet-chemical surface treatment process, all completed at room temperature and under ambient conditions. Treated substrates displayed high air stability and chemical inertia, via a reduced graphene oxide (rGO) protective composite, i.e.; the formation of AgNP-rGO composites within substrates for increased loading and enhanced active coating longevity.[53,54] The performance of these composites was compared against solely AgNP- and rGO-impregnated fabrics, with the presence of rGO acting to stabilise and facilitate the loading and adhesion of AgNP nanostructures, seemingly providing them with an improved specific surface area and subsequent improved performance.[55,56] Furthermore, leaching tests indicate enhanced and prolonged adhesion such that, even after accelerated washing cycles, the novel composites still showed high silver loading and high bactericidal efficacy over six hours of treatment, for the optimised composites. Thus, these treated textiles could impart anti-bacterial characteristics to a broad range of medical textiles, significantly reducing cross-contamination in order to lower morbidity and mortality rates as well as yield parallel reductions in hospitalisation costs.

Conclusions: This study reports on the use of reduced graphene oxide as a means of increasing overall loading and increasing the durability of adhesion of silver nanoparticles (AgNP) onto soft matter, non-woven polyviscose medical textile fabric substrates. Thus, our results show that utilisation of a simple room temperature, ambient condition synthesis method allows highly efficient loading of rGO and/or AgNP into the substrates whilst also affording a more even and improved dispersion of AgNP throughout the surface, all without the use of any stabilising agents. The seemingly synergic effects and resultant properties arising from interactions of the graphene analogue and the AgNP are the focus of continued study, with future attempts looking to parse the contributions of the various physico-chemical processes at play.

The output material allows for multiple benefits, including enhanced antimicrobial properties (i.e.; near-100% suppression of E. coli observed) with improved washfastness (i.e.; ~24% retention after 12 wash cycles) as compared with standard NP-incorporated fabrics (i.e.; <3% retention), meaning potential rinse-reuse capabilities, for enhanced effect as a non-migrating antimicrobial agent. As such, the substrates hold promise for application across a wide range of biomedical applications. More broadly, the general wet chemical technique is flexible and can readily be extended for use in other nanocomposite films and coatings on various soft matter substrates by varying the reducing agent concentration and/or NP-precursor. This surface treatment and modification process allows for reliably high loadings of functional materials containing functional groups that can efficiently bond to a substrate, to be efficiently incorporated into polymers (including metallic nanoparticles and graphene analogues for applications spanning biocidal, hydrophobic, catalytic and other applications). Thus, it offers a rapid, convenient, scalable route to conformal NP–decorated and nanocomposite soft matter coatings.

Reviewer 2 Report

The manuscript by Noor et al. describes the preparation and use of antimicrobial  of composite medical textiles.

The research work can be published with minor corrections.

1) The introduction should be shortened and more focused on the main aim of the work, moreover, it must include relevant literature data for a comparison.

2) In paragraph 2.2, AgCl was mentioned. Please mention how it was added.

3) paragraphs 2.2 and 2.3 , the sentences  "The GO...was used without further processing" and  " whilst GO was converted to rGO using ..." seems to be in contrast.

4) please discuss Ag leachant (Figure 2C, where the data are similar), in relationship to post laundering Ag content (as discussed in the text)

Author Response

The manuscript by Noor et al. describes the preparation and use of antimicrobial of composite medical textiles. The research work can be published with minor corrections.

1) The introduction should be shortened and more focused on the main aim of the work, moreover, it must include relevant literature data for a comparison.

- Agreed. The introduction has been shortened by ~200 words (please see updated section) and the focus tightened. Specific literature values have been added to the results section for the relevant data sets, for comparison, or where already present, made clearer. Please see results section in updated manuscript. (Changes have been highlighted in red text on yellow background)

2) In paragraph 2.2, AgCl was mentioned. Please mention how it was added.

- Apologies. This was an error. It should read the Ag+ instead of AgCl. The change has been made to paragraph 2.2. To be clear, there was no AgCl used as precursor in the synthesis process.

3) paragraphs 2.2 and 2.3 , the sentences  "The GO...was used without further processing" and  " whilst GO was converted to rGO using ..." seems to be in contrast.

- The meaning was that after the graphene oxide had been synthesised, there was no further chemical treatment beyond that stage, which may have otherwise affected the physico-chemical properties of the precursor material. Clarifications have been added to paragraphs 2.2 and 2.3 to avoid confusion. The change added was: The GO (5mg/ml concentration) was used in the next phase of substrate impregnation and reduction, without further processing.

4) please discuss Ag leachant (Figure 2C, where the data are similar), in relationship to post laundering Ag content (as discussed in the text)

- The explanation is a little complicated - there are several factors at play in the leaching study: 1) The amount of AgNP loading varies; 2) The amount of AgNP that is removed occurs over the course of the leaching tests; 3) The amount of AgNP material that still remains on the material after the washfastness testing.

Thus;

1) A much higher loading of silver is present on substrates (as measured by EDS) when used concurrently with rGO, as opposed to when rGO is absent;

2) The amount of AgNP material that leaches relative to this initial loading, is lower in the Ag-rGO composite, as opposed to where rGO use is absent. However, the markedly lower loading of AgNP into substrates, in the absence of rGO use means that there is a faster decrease and tailing-off of the detected silver leachant, in the solely AgNP-loaded samples, as opposed to the cases where rGO is used;

3) The variation between detected silver presence (as analysed by EDS) shows that despite the leaching values obtained, there is a lower rate of silver loss when rGO is used in combination (~75% silver loss), as opposed to when it is not (~97% silver loss). An almost order of magnitude relative difference.  

Thus, the combination of higher initial silver loading, a lower rate of relative silver leachant, as well as the much higher level of silver retention on substrates after various washfastness studies, is the explanation for the observed results.

Reviewer 3 Report

Dear Author

Please check minor spell and references. Other this is a good experimental report with multiple devices analysis. 

Author Response

We thank the reviewers for their efforts and comments. We have tried to address each of the highlighted issues in turn, giving full reasoning and references, where appropriate. We hope everything is in order.

Please check minor spell and references. Other this is a good experimental report with multiple devices analysis.

The manuscript has been double-checked and changes processed with regards to spelling, grammar, consistency and referencing issues
